# Mental health knowledge, stigma towards mental illness, and help-seeking among adolescents in secondary schools in Kampala, Uganda

Zenah Nantumbwe[1] , Rosco Kasujja[2] and Georg Schomerus[1]

[1]Psychiatry and Psychotherapy, University of Leipzig Faculty of Medicine, Germany and [2]Mental Health and Community Psychology, Makerere University, Kampala, Uganda

## Research Article

adolescents; help-seeking; mental health literacy; school-based; stigma

**Corresponding author:**
Zenah Nantumbwe;
Email: zenah.nantumbwe@medizin.
uni-leipzig.de

## Abstract

Mental health disorders are prevalent among adolescents and evidence suggests that stigma, poor mental health literacy (MHL) and access are key barriers to help-seeking for mental health difficulties in adolescence and throughout life. The study purpose is to assess existing mental health knowledge, stigma and help-seeking behaviour among adolescents in Uganda. A total of 889 secondary school students in Kampala completed standardised self-report questionnaires. The results reveal low-to-moderate levels of mental health knowledge (MAKS, range 12–60, $M = 16.35$, SD = 5.18, AMHLQ, range 33–138, $M = 64.01$, SD = 12.98), stigma (RIBS, range 4–20, $M = 12.30$, SD = 3.52) and prejudice towards people with mental illness (PPMI-TR, range 133–19, $M = 73.85$, SD = 13.38). Knowledge correlated with stigma ($r = 0.166$ and $r = 0.135$, $p < 0.01$), and with one's capacity to assess own mental health (SELF-I range 5–25, $M = 12.34$, SD = 4.4). Adolescents are open to seek help from mental health professionals but reluctant to seek it from most accessible help sources like schoolteachers. The findings provide insights for future mental health-promoting and anti-stigma interventions for adolescents.

## Impact statement

Adolescence is a critical period for developing health literacy, when early intervention for mental illnesses, and building community support structures is possible. There is a gap in understanding a nuanced and context clear picture for future school-based mental health promoting or anti-stigma interventions in low–middle-income countries with most research done in high-income regions. The study provides unique insights into adolescent population mental health knowledge, stigmatising attitudes towards mental illness and help-seeking behaviour in a distinct sociocultural, understudied world region, and low–middle-income context such as Uganda. This study found that school-going adolescents in Kampala had some knowledge about mental illness and could identify depression and anxiety and their symptoms but were less likely to identify conditions like schizophrenia and bipolar as mental illnesses. Psychotic symptoms were least understood as health concerns or illness, and stigma attitudes with prejudice and future discriminatory intentions are present among school-going adolescents including a likelihood to social distance in the future from living, working or living near someone with a mental illness, though adolescents were less likely to social distance from a friend with a mental illness. Adolescents were more open to seek help or get support from mental health professionals including school counsellors, and most unlikely from other contextually recognised help sources like senior man or woman teacher. Adolescents' knowledge and stigma levels were correlated to their capacity to self-identify as having a mental illness which impacts perceived need for help and therefore intentions to seek help. These findings highlight starting points and the need for future targeted local and regional specific anti-stigma, MHL interventions and service delivery frameworks targeting adolescents particularly in the aftermath of the COVID-19 global pandemic.

## Introduction

Adolescence is an important stage of development and is a critical time when foundational knowledge and skills on how to take care of self and health are laid down (World Health Organization, 2017). Approximately 9 million adolescents (defined as ages 10–19) make up about 25% of Uganda's population (Uganda Bureau of Statistics, 2024). According to Dick and Ferguson (2015), half of all mental health disorders onset by age 14, but most will go unrecognised and or untreated until later in life. Overall rates of mental health disorders in Uganda are reported at a prevalence of 22.9% (11.0%–34.9%) in children and 24.2% (19.8%–28.6%) in adults (Opio et al., 2022). Evidence suggests that poor mental health literacy (MHL) and stigma limit young people's access to mental health care (Erskine et al., 2015). In their qualitative research on

MHL among secondary school students in Uganda, Okello et al., 2014 point to key gaps in the knowledge and attitudes of young people that need targeted young-people-focused-interventions for improved mental health outcomes. Young people in Uganda hardly recognise symptoms of mental illness or symptoms are attributed to bad behaviour, laziness and or witchcraft (Okello et al., 2014); consequently young people are unlikely to recognise that they have a mental illness (Kigozi et al., 2010). Recognising one's own mental illness and the need for help are essential first steps in help-seeking (Schomerus et al., 2019b). Low recognition among young people reduces likelihood to engage in health-promoting actions or seeking help.

The term MHL as introduced by Jorm et al. (1997), referring to 'knowledge and beliefs about mental disorders which aid their recognition, management or prevention', is a concept that reflects a broader and positively framed public mental health goal. Low help-seeking rates are not only a problem for young individuals but also a social, economic and public health concern. By increasing MHL and early identification, it is assumed that intervention programmes could improve accessibility of mental health care at community level, delivered right where and when young people are most likely to present concerns and seek help, such as at school. Although initial research conducted on school-based MHL has been found to be effective at addressing youth mental health in other countries (Kutcher et al., 2016; Campos et al., 2018), direct intervention research in Uganda is limited with available mental health programmes and activities often targeting specific key populations or addressing adult needs.

According to current global policy frameworks, there is a need to have a population level approach to mental health promotion in across the lifespan including schools (World Health Organization, 2021). It has been found feasible to implement effective school-based mental health promotion programmes and prevention interventions in low- and middle-income countries (LMICs) that can impact the mental health and well-being of big numbers of young people (Harte and Barry, 2024). Targeted MHL interventions during or in the aftermath of a global pandemic can therefore be crucial in addressing the anticipated mental health crises or associated impacts particularly for school-aged children and adolescents (Schwartz et al., 2021).

There is limited information on studies that have utilised reliable psychometric measures to assess adolescents' MHL, knowledge, stigma attitudes and help-seeking intentions in Uganda to support implementation of targeted early interventions. The paper aims to examine status of mental health knowledge and stigma towards mental illness, and help-seeking intentions and behaviour among school-going adolescents in a distinct cultural and low- middle-income context in Uganda providing starting points for future targeted mental health promotion, stigma reduction and MHL interventions targeting adolescents.

## Methods

### Study design

The cross-sectional study is part of a larger research study on implementation and evaluation of a MHL intervention 'The Wellness Project' among adolescents in secondary schools in Kampala district, Uganda. It included three secondary schools in Kampala district, Rubaga and Kawempe division. We report baseline cross-sectional data prior to any intervention, using quantitative self-report questionnaires completed by 889 students.

### Setting

Secondary school education is delivered through both privately owned and public-funded schools. It is divided into two levels: Ordinary Level (O-Level) comprising 4 years (Senior 1–4) and Advanced Level (A-Level), 2 years (Senior 5–6). Most secondary schools have some hidden costs or tuition/school fees per school term. Schools can be either day or boarding and some are mixed sectioned (with both day and boarding section). All the three schools were in an urban area; they were mixed gender, day and boarding schools including two government-aided schools and one privately owned school.

### Ethical considerations

Ethical approval was obtained from the faculty of medicine ethics committee at the University of Leipzig 411/23- ek 090224 and with the Uganda National Council for Science and Technology through Makerere University College of Health Sciences – MAKSHSREC-2024-680. Clearance was also obtained from the Kampala Capital City Authority Directorate of Education and Welfare and from participating school administrators. Written assent was obtained from all adolescents prior to questionnaire completion, and parental or guardian consent was secured through school administrative procedures info sheets sent before the school term and filled-in parental consent forms secured before data collection.

### Sample and recruitment

The study involved ($n$ = 889) adolescents aged 10–18 years from three secondary schools in Kampala District. All students aged 10 years and above who were attending school at the time of data collection with given consent were included. Purposive sampling was used to include Universal Education Schools (USE) public schools with a population of at least ≥385 students obtained using Epi Info 7-STATCALC assuming a confidence level of 95% based on a population of 1.9 m students in secondary school in Uganda (Initiative for Social and Economic Rights, 2022). Schools were enrolled on first-response basis of school administrative approval, and entire class groups present on the data collection date were included. This non-randomised, universal school-based approach was adopted due to non-equivalence of schools and grades, as well as ethical, logistical and cost considerations like separating class groups, different school timetables and resources needed in delivering a universal intervention (Harris et al., 2006).

### Data collection

The study used anonymised paper and pen self-report questionnaires to collect data. Data were collected in classrooms between 15 June and 3 July and took approximately 45 min. Primary data were collected using standardised tools, piloted for tool usability at school C and reliability test results are included as follows:

1. Socio-demographics comprised participants' age, gender, grade/class, school and socioeconomic indicators.
2. The Adolescent Mental Health Literacy Questionnaire (AMHLQ) by Campos et al. (2016) has 33 items which measures for knowledge of mental health issues including general characteristics of mental health problems such as prevalence, signs, symptoms and risk factors for mental disorders. It also measures knowledge of three specific mental disorders including depression, anxiety and

schizophrenia/eating disorders; Responses are organised in three dimensions: knowledge/stereotypes, first aid skills and help-seeking, and self-help strategies (Campos et al., 2016; Dias et al., 2018). Schizophrenia was maintained as in original questionnaire rather than eating disorders due to social contextual relevance (α = 0.786).

3. Mental Health Knowledge Schedule (MAKS) by Evans-Lacko et al. (2010) the 12-item scale was used to measure participants' knowledge and understanding regarding stigma related to mental illness. It has been validated for use in diverse contexts.

4. Prejudice towards people with mental illness – Turkish version (PPMI-TR) adapted by Bayındır et al. (2023) originally developed by Kenny et al. (2018). The adapted Turkish version was used due to collectivist culture similarity and easier readability. The PPMI-TR 19 item scale evaluates public prejudice against people with mental illness (α = 0.605).

5. Reported and Intended Behavioural Scale (RIBS) developed by Evans-Lacko et al. (2011) measures mental health stigma-related behaviour in the general public. The RIBS analyses the presence of reported contact (items 1–4) and stigmatising or discriminatory behaviours intentions against people with mental health problems (items 5–8) in the general population (α = 0.778).

Consistent with a multidimensional conceptualisation of stigma (knowledge, attitudes and behaviour) (Thornicroft et al., 2007), we measure these components using separate validated instruments above and analysed as distinct but related constructs.

6. Self-Identification as having a Mental Illness Scale (SELF-I) by Schomerus et al. (2019a). The 5-item scale is used to assess the ability to access one's own current mental health and general susceptibility to developing a mental health problem (α = 0.626).

7. General Help-Seeking Questionnaire (GHSQ) developed by Wilson et al. (2005) is used to measure participants' intention to seek help for personal–emotional problems. The 10-item version which allows for help sources modification was adapted to match the target population context-specific help choices.

### Data analysis

Data were entered and analysed using the IBM Statistical Package for Social Sciences (IBM SPSS version 27). Exploratory data analysis was conducted on all key variables. Descriptive statistics were generated providing proportions numbers ($n$), means ($M$), standard deviation (SD), bar charts and cross tables to describe characteristics and group mean differences of participants; Pearson correlation and regression analyses were computed to find variable relationships.

### Results

#### Sample description

The sample (see Table 1) comprised $n$ = 889 secondary school students aged 10–18 years of which 477 (53.7%) of the participants were female, while 387 (43.5%) were male and 25 (2.8%) preferred not to disclose their gender. 463 students (52.1%) were between 13 and 15 years, 415 (46.7%) were 16 and 18 years, and 11 (1.2%) were aged 10–12 years. Participants were recruited from three secondary schools: enrolled in the lower secondary or Ordinary level (O-level) including three class grades.

The study sites were in Kampala, the capital city of Uganda. Socioeconomic indicators showed that most participants 76.2%

**Table 1.** Participant characteristics – demographics

| Category | Response | $n$ | (%) |
|---|---|---|---|
| Age categories | 10–12 years | 11 | 1.2 |
| | 13–15 years | 463 | 52.1 |
| | 16–18 years | 415 | 46.7 |
| Gender | Female | 477 | 53.7 |
| | Male | 387 | 43.5 |
| | Prefer not to say | 25 | 2.8 |
| School study sites | | | |
| School A | Group A | 431 | 48.5 |
| School B | Group B | 378 | 42.5 |
| School C | Group C | 80 | 9.0 |
| | Total | 889 | 100 |
| Class grade | Senior 1 | 223 | 25.1 |
| | Senior 2 | 285 | 32.1 |
| | Senior 3 | 381 | 42.9 |
| What material is the roof of the house at home? | Grass thatched/reeds | 9 | 1.0 |
| | Iron sheets | 677 | 76.2 |
| | Brick tiles | 203 | 22.8 |
| Number of rooms | 1 room | 9 | 1.0 |
| | 2–4 rooms | 172 | 19.3 |
| | 5–8 rooms | 544 | 61.2 |
| | 9–13 rooms | 154 | 17.3 |
| | 14 or more rooms | 10 | 1.1 |
| How many people stay at your home | 1–4 people | 156 | 17.6 |
| | 5–8 people | 570 | 64.2 |
| | 9–12 people | 141 | 15.9 |
| | 13 or more people | 21 | 2.4 |
| Past or current contact with someone with a mental health problem. | Yes | 589 | 66.3 |
| Contexts living or lived with; work or studied with; have or had a neighbour; or a close friend | No | 299 | 33.6 |

lived in middle-income homes with iron sheet roofing, 22.8% in high-income households with brick tile roofs and 1.9% living in low-income households with grass-thatched or reed roofs.

Most participants (570, 64.2%) lived in households with 5–8 individuals. Over half of adolescents (589, 66.3%) reported current or past contact with someone experiencing a mental health problem, answering 'Yes' to at least one of four contexts, either lived with, worked or studied with, had a neighbour or had a close friend with a mental health problem.

#### Mental health knowledge

Results in Table 2 show participants' knowledge and understanding regarding stigma related to mental illness. Depression and drug addiction as conditions were well recognised as mental illnesses by

**Table 2.** Mental Health Knowledge Schedule (MAKS) perceptions and recognition of various conditions as mental illnesses

|  | Item | Agree (%) | Neither agree nor disagree (%) | Disagree (%) | Do not know (%) |
|---|---|---|---|---|---|
| 1 | Most people with mental health problems want to have paid employment. | 35.9 | 11.7 | 12.1 | 40.3 |
| 2 | If a friend had a mental health problem, I know what advice to give them to get professional help. | 72.3 | 8.9 | 5.9 | 12.8 |
| 3 | Medication can be an effective treatment for people with mental health problems. | 61.3 | 9.2 | 21.6 | 7.9 |
| 4 | Psychotherapy can be an effective treatment for people with mental health problems. | 75.6 | 4.9 | 12.8 | 6.6 |
| 5 | People with severe mental health problems can fully recover. | 63.1 | 10.6 | 16.2 | 10.1 |
| 6 | Most people with mental health problems go to a healthcare professional to get help. | 47.2 | 9.9 | 27.0 | 15.9 |
| 7 | Depression | 76.7 | 3.9 | 11.0 | 8.3 |
| 8 | Stress | 67.0 | 8.5 | 13.7 | 10.7 |
| 9 | Schizophrenia | 35.7 | 9.2% | 3.1% | 52.0 |
| 10 | Bipolar disorder (manic-depression) | 54.1 | 8.4 | 5.6 | 31.8 |
| 11 | Drug addiction | 72.6 | 7.4 | 10.6 | 9.2 |
| 12 | Grief | 49.2 | 11.5 | 14.6 | 24.7 |

adolescents with agreement to being an illness at 76.7% and 72.6%, respectively, while schizophrenia was the least recognised condition with the highest 52.0% 'Don't know' response, and the lowest 35.7% agreement to being a mental illness. Bipolar disorder (manic-depression) was also a less recognised condition with 31.8% of participants selecting 'Don't know' if it is a mental illness.

There was high agreement on treatment effectiveness with most adolescents (75.6%) agreeing that psychotherapy and medication (61.3%) can be an effective treatment for people with a mental health problem.

The results in Table 3 show specific MHL domains including knowledge of mental health problems, erroneous beliefs or stereotypes, first aid skills and help-seeking, self-help strategies. The total AMHLQ score mean average for adolescents was 64.01 (SD = 12.98). The subscale self-help strategies had the lowest average score at 7.55 (SD = 2.90) out of 24.

Results in Supplementary Table S1 provide more details of the AMHLQ. Most adolescents recognised the contribution of lifestyle factors to good mental health, including sleeping well (74.7%), physical exercise (72.2%) and a balanced diet (65.0%), while 58.5% identified alcohol use as a potential risk factor. Adolescents' cognitive behavioural understanding was strong, with 93.2% agreeing that mental disorders affect thoughts, and

74.5% rejecting that they do not influence behaviour or emotions (68.3%).

There are marked differences in participants' MHL across specific disorders. Knowledge of depression, anxiety and substance-use disorders and their symptoms was relatively high and more consistent, while knowledge of schizophrenia including hallmark psychotic symptoms was limited. Recognition of depressive symptoms such as person feels very miserable (89.0%) and loss of interest or pleasure (80.6%) was strong, as was knowledge of anxiety-related panic in feared situations (84.8%) and avoidance of distressing situations (67.6%). Less than half of participants identified hallucinations (42.1%) and delusions (34.8%) as symptoms of schizophrenia.

First aid skills and help-seeking intentions towards others were generally positive. Most participants (83.9%) reported they would listen supportively to someone with a mental disorder. Adolescents were likely to encourage professional help-seeking for a friend or relative from psychologists/psychiatrists (84.5%) or doctors (80.3%), and 77.9% would seek such help themselves if they had a mental disorder. Fewer participants would seek support from relatives (43.7%), friends (44.8%) and if a friend developed a mental disorder they would talk to a teacher (52.3%).

### Stigma towards mental illness

Results presented in Table 4 RIBS items 5–8 related to participants' future intended behaviour likelihood to social distance, from someone with a mental illness in different contexts. Approximately half (50.6%) reported unwillingness to live with, and 46.9% expressed reluctance to live near someone with a mental health problem. Similar patterns were observed in workplace settings, with 45.0% disagreeing that they would like to work with someone with a mental illness.

Continuing a relationship with a friend who developed a mental health problem showed the least intention to social distance in the future, 53.7% of participants stated they would continue a relationship with a friend who developed a mental health problem. Notably, 28.0% expressed disagreement and 9.3% uncertainty.

**Table 3.** Adolescent Mental Health Literacy Questionnaire (AMHLQ)

|  | Mean | SD | Minimum score | Scale maximum |
|---|---|---|---|---|
| AMHLQ Total score | 64.01 | 12.98 | 33.00 | 138 |
| AMHLQ Erroneous beliefs/ Stereotypes | 20.42 | 5.65 | 10.00 | 49 |
| AMHLQ Knowledge of mental health problems | 21.23 | 5.60 | 11.00 | 50 |
| AMHLQ First aid skills and help-seeking behaviour | 14.82 | 4.90 | 7.00 | 35 |
| AMHLQ Self-help strategies | 7.55 | 2.90 | 5.00 | 24 |

**Table 4.** Reported Intended Behavioural Scale (RIBS) social distancing

| RIBS statement | Agree (%) | Neither agree nor disagree (%) | Disagree (%) | Do not know (%) |
|---|---|---|---|---|
| 5. In the future, I would be willing to live with someone with a mental health problem. | 20.5 | 12.6 | 50.6 | 16.3 |
| 6. In the future, I would be willing to work with someone with a mental health problem. | 26.1 | 14.5 | 45.0 | 14.4 |
| 7. In the future, I would be willing to live nearby to someone with a mental health problem. | 23.3 | 14.7 | 46.9 | 15.1 |
| 8. In the future, I would be willing to continue a relationship with a friend who developed a mental health problem. | 53.7 | 9.0 | 28.0 | 9.3 |

Results in Supplementary Table S2 present on the PPMI-TR scale, across domains such as unpredictability, fear and avoidance, malevolence, and authoritarianism.

Most participants viewed people with mental illness as likely to do unexpected things (81.2%) and that their behaviour is unpredictable (77.0%). Regarding fear and avoidance, 61.3% agreed, they would find it hard to talk to someone with a mental illness, although 57.5% said they would not be scared of a person with mental illness. While 22.8% agreed it is best to avoid people with mental illness, 61.6% disagreed, with most adolescents rejecting avoidance-based attitudes.

In the malevolence domain, 64.1% agreed that people with mental illness avoid life's difficulties, and 32.7% viewed them as genetically inferior. However, most adolescents 79.1% disagreed that people with mental illness do not deserve sympathy.

Concerning authoritarianism and autonomy, 52.7% disagreed that people with mental illness should be free to make their own decisions, yet 57.6% agreed they should be allowed to live as they wish, and 51.2% agreed society has no right to limit their freedom.

### Self-identification as having mental illness

Results on self-identification as having a mental illness (SELF-I) (see Supplementary Table S3) reveal that on personal assessment of their own mental health, 64.3% adolescents agreed that they are mentally stable and do not have a mental health problem, and at least half of adolescents 55.5% see themselves as mentally healthy and emotionally stable. Over 61.8% of respondents disagree that they could be the type of person likely to have a mental illness. While 19.9% agreed, at least half the adolescents 54.2% do not believe their current issues are the first signs of a mental illness, about 40.2% of adolescents are doubtful about the idea of themselves having a mental illness.

### Help-seeking

Results in Table 5 reflect adolescents reported likelihood to seek help from sources listed for personal or emotional problems. Parents or guardians were the most endorsed source for help (68.9%), mental health professionals were also highly endorsed by adolescents (66.8%), followed by doctors or nurses (55.6%), and ministers or religious leaders (50.6%). Teacher (i.e., class teacher, senior female or male teacher) was the most unlikely to be approached (63.4%).

In open-ended responses on question 10, adolescents mostly listed parental figures names, specific friends, culturally relevant sources like traditional healers, and digital resources such as apps, social media and most notably artificial intelligence (AI) tools Snapchat AI, Meta-AI or AI in general. Although 70.6% were

**Table 5.** General Help-Seeking Questionnaire (GHSQ) likelihood of seeking help from different sources for personal or emotional problems

| Help source | Unlikely (%) | Neither unlikely nor likely (%) | Likely (%) |
|---|---|---|---|
| 1. Friend (not related to you) | 54.2 | 3.1 | 42.6 |
| 2. Intimate partner (e.g., girlfriend, boyfriend) | 45.0 | 3.9 | 51.0 |
| 3. Parent/guardian | 27.8 | 3.1 | 68.9 |
| 4. Other relative/family member | 46.2 | 5.3 | 48.4 |
| 5. Mental health professional (e.g., psychologist, social worker, counsellor) | 28.5 | 4.5 | 66.8 |
| 6. Teacher (e.g., Class teacher, senior female or male teacher) | 63.4 | 6.1 | 30.6 |
| 7. Doctor/nurse | 37.8 | 6.6 | 55.6 |
| 8. Minister or religious leader (e.g., Priest, Sheikh, Pastor) | 44.7 | 4.5 | 50.6 |
| 9. I would not seek help from anyone | 70.6 | 3.7 | 25.6 |
| 10. I would seek help from another not listed above (please list in the space provided, e.g., Traditional healer, phone helpline or app (If no, leave blank)) ____________________ | | | |

willing to seek help from at least one source, 25.6% reported they would not seek help from anyone.

### Correlations

Analysis done using Pearson correlations in Supplementary Table S4 revealed several statistically significant relationships between knowledge or MHL, stigma attitudes, and stigma-related variables. A positive correlation was observed between the MAKS and AMHLQ scores, $r = 0.290$, $p < 0.01$. Both scales measuring mental health knowledge, were associated with higher scores in specific MHL domains in the AMHLQ among adolescents. Both MAKS and AMHLQ scores were positively correlated with RIBS scores, $r = 0.166$ and $r = 0.135$, respectively, significant; it reveals a relationship that the higher the knowledge or MHL, the more accepting and less likelihood to social distance from people with a mental illness.

MAKS and AMHLQ scores were also negatively correlated with Prejudice towards PPMI-TR, $r = -0.106$ and $r = -0.071$, respectively. These results imply that higher scores in knowledge and MHL are linked with lower prejudicial attitudes. The RIBS score showed a negative correlation with the PPMI-TR score, $r = -0.428$, $p < 0.001$,

indicating that lower PPMI-TR scores which reflect higher prejudice were strongly associated with higher future likelihood to social distance. Correlations found between Self-Identification SELF-I and variables, including AMHLQ ($r = 0.071$, $p < 0.05$), showed the higher knowledge was associated with a higher capacity to spontaneously access current mental health status and self-identify as having a mental illness. The negative correlation with RIBS ($r = -0.116$, $p < 0.001$) indicating higher capacity to self-identify as having a mental illness is associated with less likelihood to social distance in the future while a positive correlation with PPMI-TR ($r = 0.085$, $p < 0.05$), higher scores on the PPMI-TR which indicate less prejudicial attitudes were associated with higher capacity to self-identify as having a mental illness.

## Discussion

Mental health knowledge or literacy is widely recognised as essential for reducing stigma and promoting help-seeking; however, the specific literacy dimensions requiring targeted intervention remain unclear. In one of the most culturally heterogeneous countries in Sub-Saharan Africa, with diversity by ethnicity, religion, healing practises and other social factors, we explore what knowledge, stigma towards mental illness and help-seeking behaviour exists among school-going adolescents in an East African capital city using standardised tools developed in the global north. The present study highlights the need for nuanced, context-specific approaches that consider sociocultural conceptualisation of mental illness and regional factors when assessing and improving adolescent literacy.

Participants demonstrated moderate literacy, with relatively strong knowledge of common conditions such as depression, anxiety and substance use, and a general openness to professional help-seeking. In contrast, knowledge of psychotic disorders and symptom recognition was low, accompanied by stigma attitudes with prejudice, social distancing from people with mental illness and with limited self-help-seeking strategies. Mean scores on both the MAKS and AMHLQ were comparatively low, with an average AMHLQ score of 64.01 (SD = 12.98) out of 138, lower than ($M = 72.59$, SD = 25.26) as reported by Zare et al. (2022) among Iranian students. The findings indicate overall low-to-moderate MHL among adolescents, with notable knowledge differences across literacy dimensions and disorders.

While participants had moderate literacy and symptom recognition for common mood and anxiety disorders, there was a marked gap in recognising more stigmatised conditions such as schizophrenia and bipolar (manic-depression) disorder. In comparison, Ben Amor et al. (2023) reported that 85.6% of Tunisian students recognised schizophrenia as a mental illness, with only 3.5% unsure, and Evans-Lacko et al. (2010) found similar results among young adults in London (78.2% recognition; 8% unsure). In contrast, our study showed substantial differences in knowledge across disorders, with schizophrenia being the least recognised condition: only 35.7% of adolescents recognised it as a mental illness, while 52.0% reported 'don't know'. Likewise, 31.8% were 'don't know' regarding whether bipolar (manic-depressive) disorder was a mental illness. Such difference in the findings between conditions may also reflect broader societal patterns in mental health knowledge of disorders, with schizophrenia remaining one of the least understood and most discriminated against or stigmatised condition worldwide (Thornicroft et al., 2009).

Adolescents rarely identified hallmark psychotic symptoms such as delusions and hallucinations as signs of illness. This may

reflect that 'idioms of distress' for mental illness in Uganda which contain locally and culturally shared interpretations, significance and relevance of social, psychological and somatic symptoms; Asiimwe et al. (2023) differ from the Western biomedical conceptualisation of distress symptoms which the questionnaires utilised. Our findings on psychotic disorder literacy align with qualitative research in Uganda showing that conditions such as bipolar, schizophrenia, dissociative disorders and psychotic depression are often perceived as an outcome of broken associations with ancestors, that is, clan problems, witchcraft (Abbo, 2011; Van Duijl et al., 2014) and in other cases as a sin against a Holy God (Verginer and Juen, 2019). These culturally grounded meanings shape what is considered abnormal and requiring treatment and could explain regional differences in literacy of specific disorder symptoms highlighting the need for targeted education addressing gaps in understanding severe mental illness that incorporates sociocultural conceptualisation of illness.

Consistent with trends reported in the United Kingdom between 2009 and 2019, where public endorsement of stress and grief as mental illnesses increased (Henderson, 2023), the present study found that adolescents also frequently perceived stress and grief as mental disorders. Although these are not clinical conditions, this pattern highlights the need for clearer public messaging distinguishing emotional distress from diagnosable or clinical mental illness. While recent mental health awareness efforts in media at population level appear effective at increasing general recognition, future initiatives should focus on improving disorder-specific understanding and clarified information to lessen concept creep.

Recent public anti-stigma campaigns have largely focused on general mental health problems, barely covering specific disorders and avoiding discussion of severe disorders like psychosis due to fear of public response (Henderson, 2023). In contexts such as Uganda where aetiological understanding of mental illness includes witchcraft, curses, clan problems, bad behaviour, with limited recognition of signs and symptoms of the mental health disorders (Okello et al., 2014), excluding severe disorders from public messaging may be detrimental to efforts of improving MHL and reducing stigma. It is even more critical to include and talk about severe or psychotic disorders, symptoms and appropriate help-seeking and behaviour towards others. Mental health promotion and anti-stigma initiatives should integrate information including severe psychotic disorders like schizophrenia (Henderson, 2023) and bipolar to counteract stereotypes before they take shape.

Knowledge gaps, stigma and prejudice were evident in stereotypical misconceptions and scepticism regarding recovery and employment intentions among people with mental illness. This may explain high stigma, particularly for severe or psychotic disorders. Public awareness and MHL interventions should provide information on a mental health-illness continuum to improve attitudes towards people with mental illness (Schomerus et al., 2016) while reinforcing confidence in established biomedical treatment modalities and recognising cultural explanatory models.

Stigma-related future discriminatory behaviour or social distancing was evident, with half (50.6%) of the adolescents reporting discomfort or unwillingness to live with someone with a mental illness, compared to 25.8% in a study among Tunisian students by Ben Amor et al. (2023) showing a higher degree of future social rejection or discriminatory behavioural intent towards people with mental illness among our participants. The limited understanding of some conditions and their symptoms could explain continued prejudice and social distancing. Educational workshops targeting

adolescents shown to produce positive changes in stigma attitudes and help-seeking (Pinfold et al., 2003) remain critical.

In our study, continuing a relationship with a friend who developed a mental health problem showed the least intention to social distance in the future across scenarios, with 62.7% of adolescents reported willingness to maintain the friendship. This proportion, although lower than 83.7% reported by Ben Amor et al. (2023) among students, suggests that adolescents may support existing friends with mental health conditions. Personal connections, as noted by Ronaldson and Henderson (2024), may foster more support for non-discrimination against individuals with mental illnesses compared to other scenarios. This highlights the need for holistic stigma-reduction interventions incorporating empathy-building, openness about lived experience narratives and critical reflection. Combining education with strategies such as contact-based interventions or peer discussions (Schulze et al., 2003; Thornicroft et al., 2016) is likely essential to achieve deeper and lasting attitudinal change.

Adolescents demonstrated moderate knowledge and strong recognition of professional mental health treatments with biomedical interventions like psychotherapy and medication seen as effective and expressed willingness to support peers experiencing difficulties. They were most likely to seek help from parents and mental health professionals such as counsellors, psychologists and psychiatrists. The preference for mental health professionals suggests a growing acceptance of formal or professional mental health care in Uganda among young people.

Participants were more sceptical about seeking support from relatives, peers and school figures, especially teachers despite teachers being the most accessible sources of help, this particularly noteworthy for boarding school students who spend extended periods at school with three terms each about 3 months long annually. This reluctance may reflect discomfort or stigma surrounding school-based informal support channels like teachers (senior man or woman teachers). Although national policies (Ministry of Health 2017; Ministry of Education and Sports Uganda 2020) of the mental health identify peer networks and school staff including guidelines which recommend that schools have senior female or male teachers, as key resources for addressing adolescent emotional and mental health needs, these findings suggest such resources remain underutilised. Future qualitative and interventional research could explore barriers to engaging these most available school-based help sources.

There is a need for greater collaboration between the formal mental health system and 'informal mental health providers' like religious leaders, faith leaders and traditional healers as they were identified by adolescents as a likely support resource and considering they are often more accessible within communities and religion and spirituality are social–cultural backbones of how mental illness is experienced, expressed and responded to in Uganda (Asiimwe et al., 2023). Such collaboration efforts as recommended by Dwanyen et al. (2024) with various stakeholders in Uganda like these identified figures may strengthen referral pathways and structures in low resource settings.

Additionally, adolescents noted technological and digital resources, including apps and AI tools integrated into social media platforms, as potential mental health supports. Given the growing reliance on emerging AI or digital tools, there is a critical need for evidence-based evaluations of the capacity, safety and effectiveness of these emerging interventions for adolescent population particularly in low–middle-income contexts where digital resources may be relied on to play a pivotal role to change the situation with current challenges to access professional mental health services.

Respondents scored particularly low on self-help strategies subscale with mean of 7.55 out of 24, compared to mean of 10.71 among Iranian students (Zare et al., 2022), indicating low confidence in using personal strategies to manage mental health. The lower scores alongside low beliefs about recovery highlight the need for initiatives that both inform and challenge stereotypes while encouraging proactive, supportive, positive health behaviours within communities.

The notable differences in literacy and attitudinal scores between Ugandan adolescents in comparison with peers across different world regions highlight the need for context-specific understanding of stigma and MHL across regions as proposed by Bhavsar et al. (2019) that different populations could be in particular need of anti-stigma and literacy interventions as differences could be due to various reasons within the regions.

## Limitations

The study has some inherent limitations. It relied on self-reported data which may be influenced by social desirability bias. It is not applicable to generalise these findings across the country or region, because Uganda exhibits multicultural diversity and varied socio-demographic backgrounds including type of schools. The study was conducted among urban adolescents studying and living in the capital city and did not address adolescents living in rural areas. The used tools were not validated for Ugandan context. The study results here are quantitative and lack the qualitative aspect of how and why these findings were observed.

## Recommendations for practice and policy

The study highlights the importance of enhancing MHL among adolescents and young adults to reduce stigma and promote early help-seeking. To support this, schools, parents, governing structures, researchers and youth organisations need to consider the regional social–cultural differences in knowledge and stigma about different mental illnesses and during designing and implementation of mental health interventions. These model interventions should not only address common disorders like depression and anxiety but also incorporate information on less understood or highly stigmatised psychotic conditions during interventions among adolescents. Researchers and organisations in public health and stigma prevention could consider proposing a streamlined or curriculum-based MHL information that does not exclude the most misunderstood, stigmatised disorders, particularly for children and adolescent populations to counter stereotypes early. Furthermore, effective intervention programmes should be grounded in clear understanding of adolescents' sociocultural contexts. This includes recognising sociocultural interpretations of symptoms of mental illness and help-seeking preferences.

## Conclusion

In conclusion, this paper builds on the limited growing body of literature in Uganda on adolescent MHL. We provide evidence of current adolescent population knowledge levels, existing stigma attitudes, help-seeking sources preferred among adolescents in Uganda. We found low-to-moderate levels of knowledge, among

school-going adolescents, strong stigma towards people with mental illness with some prejudice and future discriminatory behavioural intentions. With distinctly lower knowledge for psychotic conditions *versus* common disorders and an openness to seek help from parental figures, mental health professionals including counsellors and least likely from teachers like senior man and woman teachers. MHL interventions should consider differences in sociocultural experiences which play a part in understanding mental illness and stigma when working among adolescents taking time to target and clarify areas with low knowledge with consideration of cultural knowledge and understanding.

## Abbreviations

| | |
|---|---|
| MHL | mental health literacy |
| LMICs | low–middle-income countries |
| MAKS | Mental Health Knowledge Schedule |
| AMHLQ | Adolescent Mental Health Literacy Questionnaire |
| RIBS | Reported and Intended Behavioural Scale |
| PPMI-TR | Prejudice towards People with Mental Illness scale Turkish version |
| GHSQ | General Help-Seeking Questionnaire |

**Open peer review.** To view the open peer review materials for this article, please visit http://doi.org/10.1017/gmh.2026.10194.

**Supplementary material.** The supplementary material for this article can be found at http://doi.org/10.1017/gmh.2026.10194.

**Data availability statement.** The data that support the findings of this study are available through the corresponding author, [Z.N], upon reasonable request. Restrictions apply to the availability of these data. All data of the Wellness Project are the property of the Faculty of Medicine, University of Leipzig and are subject to the Law for the Protection of Informal Self-Determination in the Free State of Saxony (Saxon Data Protection Act).

**Acknowledgement.** We thank all the members of the Wellness Psychological Services team including Yvonne Zabu and those, who were involved in the study conduct and data collection Dr. Simon Kizito from Makerere University and all the participants who took part in the Wellness Project Study. We also acknowledge support from Leipzig University within the program of Open Access Publishing.

**Author contribution.** Conceptualisation: Z.N.; Data curation: Z.N.; Formal analysis: Z.N.; Funding acquisition: Z.N., G.S.; Investigation: Z.N.; Project administration: Z.N.; Resources: Z.N., R.K., G.S.; Supervision: G.S.; Validation: G.S.; Visualisation: Z.N.; Writing – original draft preparation: Z.N.; Writing – review and editing: Z.N., G.S., R.K. All authors have read and agreed to the published version of the manuscript.

**Financial support.** This research is supported a research assistantship and material costs by the Medical Faculty of the University of Leipzig and by Wellness Psychological Services Uganda with in-kind infrastructural support.

**Competing interests.** The authors declare no competing interests.

**Ethics approval statement.** Ethical approval to conduct The Wellness Project Study was conducted according to the guidelines of the Declaration of Helsinki and approved by the Ethics Committee of the Medical Faculty at the University of Leipzig (registration number: 411/23- ek 090224, date of approval 09.02.2024) and with the Uganda National Council for Science and Technology through Makerere University College of Health Sciences – MAKSHSREC-2024-680. Date of approval 09.04.2024.

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
