## [Reviewer Report]

Thank you for the opportunity to review this interesting investigation of young people’s knowledge and attitudes towards mental health, and their help-seeking behaviour using a number of questionnaire-based measurements, nested as the baseline of a wider study.

1. Lines 33-35; check sentence ‘Adolescents’ knowledge and stigma levels were correlated to their capacity to self-identify as having a mental illness and therefore intentions to seek help.’. Are knowledge and stigma levels both correlated positively with capacity to self-identify? And is self-identifying so simply linked to seeking help?

2. This topic needs more critical interrogation in the introduction also (lines ~55-61). What is the exact nature of the link between knowledge/MH literacy and subsequent help-seeking?

3. More detail is needed in ethics on assent. How were they engaged and given info. Did parents consent?

4. Data collection; were questionnaires self-completed or with a researcher? In what setting?

5. Aside from 4 (PPMI-TR) no comment is made on local translation/adaptation/testing procedure and the justification for these measures and (presumed) validity for the setting. RIBS is a measure of potential discriminatory behaviour (enacted stigma) as directly measuring discriminatory behaviour is challenging.

6. The results are clearly described, and generally in keeping with expectations.

7. Discussion;

- Line 308-309 states that nuanced, context-specific approaches are needed. What do you mean here? What could be done better in this study for example?

- It was interesting to see schizophrenia and BPD recognition so low, given that they are often seen as archetypal mental conditions, where depression and anxiety symptoms are sometimes interpreted as closer to normal experience. The potential explanation for this is useful.

- More nuance is needed in discussing how to address the current perceptions of causation as spiritual. While increasing a population’s understanding of the potential benefits of seeking orthodox health care might increase formal health seeking, there is mixed evidence on the degree to which medicalising cause reduces stigma.

It was interesting to see high levels of reporting seeking help from professionals.

- It was good to see the conclusion related to working with traditional healers, but this could be linked to better understanding what impact there is on challenging traditional causation as an anti-stigma approach.

- There could have been more comparisons with other African studies as most related to the UK or Iran.

8. Limitations; It should have been noted that there is a high risk of social desirability bias in any assessment of attitudes where ‘acceptable’ opinions might be easily guessed by participants. Also that the high number of statistical tests (particularly where sub-domains are included in separate analyses) carried out makes it more likely that associations are found by chance.

There are a number of typos and phrasing issues to address:

- line 22; stigma attitudes should probably be stigmatising attitudes

- line 25; mental health- illness should probably be mental illness, or mental ill health

- line 33; senior man or woman teacher perhaps better as senior male or female teacher

- line 47; ‘take care of self and own health’ needs rewriting

- line 49; According to (Dick and Ferguson 2015) is not aligned to referencing style. Same in line 162.

- line 74; ‘...to have A population-wide...’

- line 82; ‘There’s is too informal. Use ’There is'.

- line 82; What does ‘inaccessible information’ mean? please clarify

- line 119; please review sentence; ‘Parental or guardian consent for adolescent participation in study was obtained through school administrative structures before of the school term.’

- lines 185-189; please review the sentence as is long and confusing

- line 261; please clarify ‘...social context specific sources like traditional healers,..’

Overall these findings are useful, but more nuance could have been achieved (as the authors say) based on locally-designed measures or qualitative approaches.

---

## [Reviewer Report]

Major Comment:

The investigators should clarify their conceptualization of the construct of stigma. There is general consensus in the mental health literature that stigma is conceptualized as deficits in knowledge, poor attitudes towards people with mental ill health and as discriminatory behavior. The present study has conceptualized stigma as discriminatory behavior only (measured by RIBS) and have measured knowledge as a separate construct (using MAKS). Please provide justification for this.

All used tools were not validated for the Uganda context. Discuss this limitation.

Methods

- From the description of the selection criteria of the schools that were included in the study, this was a convenient sample selection and not purposive sample. For example, the authors recruited schools based on access and obtaining permission from the school administration.

- To help readers understand how representative (or otherwise ) the sample is, provide examples of the ethical, logistical and cost considerations mentioned in lines 131-132 that varied between included schools and those that qualified but were not included

- Describe in more detail the consent process. For children under the age of 18, was assent sought?

Results

- The authors mentioned that reliability of the scales was evaluated but these results are not presented here

Discussion

- In the recommendations for policy and practice, authors should align these with their findings. For example, their study shows stark differences between awareness of depression and anxiety and that of psychosis. What would a model intervention look like?